# REWRITER-EVALUATOR FRAMEWORK FOR NEURAL MACHINE TRANSLATION

## ABSTRACT

Encoder-decoder architecture has been widely used in neural machine translation (NMT). A few methods have been proposed to improve it with multiple passes of decoding. However, their full potential is limited by a lack of appropriate termination policy. To address this issue, we present a novel framework, *Rewriter-Evaluator*. It consists of a rewriter and an evaluator. Translating a source sentence involves multiple passes. At every pass, the rewriter produces a new translation to improve the past translation and the evaluator estimates the translation quality to decide whether to terminate the rewriting process. We also propose a prioritized gradient descent (PGD) method that facilitates training the rewriter and the evaluator jointly. Though incurring multiple passes of decoding, *Rewriter-Evaluator* with the proposed PGD method can be trained with similar time to that of training encoder-decoder models. We apply the proposed framework to improve the general NMT models (e.g., Transformer). We conduct extensive experiments on two translation tasks, Chinese-English and English-German, and show that the proposed framework notably improves the performances of NMT models and significantly outperforms previous baselines.

## 1 INTRODUCTION

Encoder-decoder architecture (Sutskever et al., 2014) has been extensively used in neural machine translation (NMT) (Vaswani et al., 2017; Zhang et al., 2019). Given a source sentence, an encoder firstly converts it into hidden representations, which are then conditioned by a decoder to generate the target sentence. Attention mechanism (Bahdanau et al., 2015) is very effective in learning the alignment between a source sentence and a target sentence. Hence, attention mechanism is usually used in the architecture to improve its capability, such as capturing long-distance dependencies.

Similar to traditional machine learning efforts (Zhang & Gildea, 2008), some recent approaches in deep learning attempt to improve encoder-decoder architecture with multiple passes of decoding (Xia et al., 2017; Zhang et al., 2018; Geng et al., 2018). NMT refers this to polish mechanism (Niehues et al., 2016). Under this scheme, more than one translations are generated for a source sentence and, except for the first translation, each of them is based on the translation from the previous decoding pass. While these methods have achieved promising results, they lack a proper termination policy to the multi-turn process. Xia et al. (2017); Zhang et al. (2018) adopt a fixed number of decoding passes that can be inflexible in deciding the optimal number of decoding passes. Geng et al. (2018) use reinforcement learning (RL) (Sutton et al., 2000) to automatically decide the optimal number of decoding passes. However, RL is unstable due to its high variance of gradient estimation and objective instability (Boyan & Moore, 1995). Since these methods may have premature termination or over translation, their potential can be limited.

To address this problem, we propose a novel framework, *Rewriter-Evaluator*, in this paper. It consists of a rewriter and an evaluator. The translation process involves multiple passes. Given a source sentence, at every pass, the rewriter generates a new target sequence aiming at improving the translation from prior passes, and the evaluator measures the translation quality to determine whether to terminate the rewriting process. We also propose a prioritized gradient descent (PGD) method that facilitates training the rewriter and the evaluator jointly. The essential idea is using a priority queue to improve sampling efficiency by collecting the translation cases that yield low scores from the evaluator for next-pass rewriting. The size of the queue is a few times larger than the batch size.

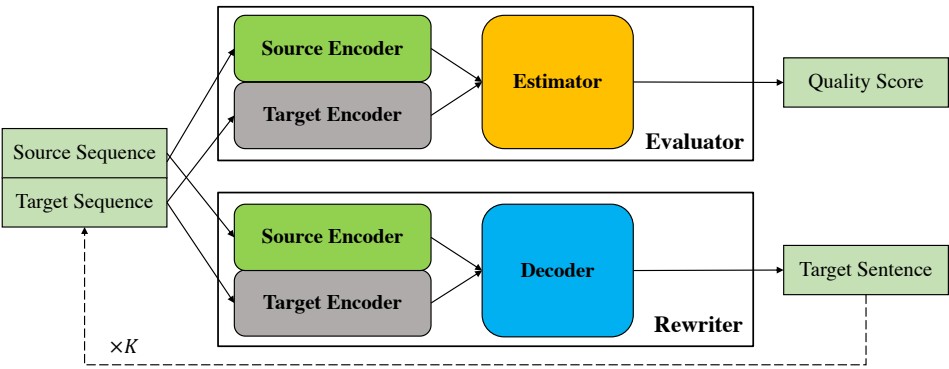

Figure 1: This illustrates the architecture of the propose *Rewriter-Evaluator*.

Although *Rewriter-Evaluator* involves multiple decoding passes, training time using PGD method is comparable to that of training an encoder-decoder (Bottou & Bousquet, 2008) that doesn't have multiple decoding passes.

We apply *Rewriter-Evaluator* to improve the widely used NMT models, RNNSearch (Bahdanau et al., 2015) and Transformer (Vaswani et al., 2017). Extensive experiments have been conducted on two translation tasks, Chinese-English and English-German, to verify the proposed method. The results demonstrate that the proposed framework notably improves the performance of NMT models and significantly outperforms prior methods.

## 2 REWRITER-EVALUATOR

### 2.1 ARCHITECTURE

The architecture of *Rewriter-Evaluator* is shown in Figure 2. Our framework consists of a rewriter and an evaluator. The process of translating a source sentence $\mathbf{x}$ consists of multiple passes. At each iteration $k \geq 1$, assuming the translation from the prior iteration $k - 1$ is $\mathbf{z}^{(k-1)}$, the rewriter produces a new translation $\mathbf{z}^{(k)}$ and the evaluator estimates its quality score $q^{(k)}$. Formally, the $k$-th iteration of a translation process is defined as

$$\mathbf{z}^{(k)} = \text{rewriter}(\mathbf{x}, \mathbf{z}^{(k-1)}), \quad q^{(k)} = \text{evaluator}(\mathbf{x}, \mathbf{z}^{(k)}). \tag{1}$$

Initially, $\mathbf{z}^{(0)}$ and $q^{(0)}$ are respectively set as an empty string and a large negative number.

In the following, we describe an architecture that shares the sentence encoders between rewriter and evaluator. However, the proposed method is not restricted to this particular architecture.

At the $k$-th pass, the source sequence $\mathbf{x}$ and the translation $\mathbf{z}^{(k-1)}$ from the previous pass are respectively embedded by a source encoder $f^{SE}$ and a target encoder $f^{TE}$:

$$\mathbf{h} = f^{SE}(\mathbf{x}), \quad \mathbf{p}^{(k-1)} = f^{TE}(\mathbf{z}^{(k-1)}). \tag{2}$$

The rewriter $\psi$ contains a decoder $g^{DEC}$ to produce a new translation $\mathbf{z}^{(k)}$:

$$\mathbf{z}^{(k)} = \psi(\mathbf{x}, \mathbf{z}^{(k-1)}) = g^{DEC}(\mathbf{h}, \mathbf{p}^{(k-1)}), \tag{3}$$

where $g^{DEC}$ can be any conditional language model (e.g., Transformer).

The evaluator $\phi$ measures the translation quality with an estimator $g^{EST}$ as

$$q^{(k)} = \phi(\mathbf{x}, \mathbf{z}^{(k)}) = g^{EST}(\mathbf{h}, \mathbf{p}^{(k)}). \tag{4}$$

Here the estimator can be any text matching model, e.g., ESIM (Chen et al., 2016).

The above procedure is repeatedly carried out until the following condition is satisfied:

$$q^{(k)} + \delta < q^{(k-1)}, \delta > 0, \tag{5}$$

or a certain number of iterations $K > 0$ is reached. In the former case, we use $\mathbf{z}^{(k-1)}$ as the final translation. In the latter case, the last translation $\mathbf{z}^{(K)}$ is accepted.

---

**Algorithm 1:** Optimization via Prioritized Gradient Descent

---

**Input:** Rewriter $\psi$; evaluator $\phi$; training set $T$; batch size $B$; expected iterations $E$.
**Output:** Well trained rewriter $\psi$; well trained evaluator $\phi$.

1  Initialize an empty priority queue $A$ with capacity $C = B \times E$.
2  **while** *Models are not converged* **do**
3      Randomly sample a $B$-sized batch of training samples $S$ from $T$.
4      **for** $(\mathbf{x}, \mathbf{y}) \in S$ **do**
5          Push quadruple $(\mathbf{x}, \mathbf{y}, [\text{"SOS"}, \text{"EOS"}], -\infty)$ into queue $A$.
6      Pop the samples that exceed that the capacity from queue $A$.
7      Initialize an empty priority queue $D$ of limited size $C$.
8      Initialize an empty list $F$ to collect samples for training.
9      **for** $(\mathbf{x}, \mathbf{y}, \mathbf{z}^{(k-1)}, r^{(k-1)}) \in A$ **do**
10          Obtain rewrite $\mathbf{z}^{(k)}$ and evaluation score $q^{(k)}$, respectively, using Eqs. (3) and (4).
11          Add sample $(\mathbf{x}, \mathbf{y}, \mathbf{z}^{(k-1)}, q^{(k)})$ into list $F$.
12          Compute new quality score $r^k$ using Eq. (8).
13          Push quadruple $(\mathbf{x}, \mathbf{y}, \mathbf{z}^{(k)}, r^{(k)})$ into queue $D$.
14      Update rewriter $\psi$ with the samples in list $F$ using Eq. (6).
15      Update evaluator $\phi$ with the samples in list $F$ using Eq. (7).
16      $A \leftarrow D$.

---

## 2.2 TRAINING CRITERIA

The rewriter $\psi$ is trained with teacher forcing. To learn generating word at position $i$, it uses ground truth of the prior time steps $[y_0, \cdots, y_{i-1}]$ as follows

$$\pi_i = \psi(\mathbf{x}, \mathbf{z}^{(k-1)}, [y_0, y_1, \cdots, y_{i-1}]), \quad \mathcal{J}^\phi = \sum_{1 \leq i \leq N} -\log(\pi_i[y_i]), \qquad (6)$$

where $N$ is the length of $\mathbf{y}$. The first symbol $y_0$ and the last symbol $y_N$ are "SOS" for sentence beginning and "EOS" for sentence ending, respectively.

For evaluator $\phi$, when presented with the ground truth $\mathbf{y}$ and the predicted quality of current translation from Eq. (4), it incurs a hinge loss as

$$q^* = \phi(\mathbf{x}, \mathbf{y}), \quad \mathcal{J}^\psi = \max(0, 1 - q^* + q^{(k)}). \qquad (7)$$

The translation $q^{(k)}$ can be generated via greedy search or beam search.

## 3 PRIORITIZED GRADIENT DESCENT METHOD

We introduce, in Algorithm 1, a prioritized gradient descent (PGD) method to bias the training towards rewriting poor translations with certain quality scores. Different from random sampling from the whole training set in the stochastic gradient descent (SGD), it uses a priority queue to keep poorly translated cases for sampling in each mini-batch of the training.

The procedure starts with an empty priority queue (1-st line) with capacity $C$ no greater than the product of mini-batch size $B$ and expected iterations $E$ (i.e., $C \leq B \times E$). The priority queue is ordered with a quality score in ascending order with the top one corresponding to the lowest score. The quality score of a certain sample $(\mathbf{x}, \mathbf{y}, \mathbf{z}^{(k)})$ is computed as

$$r^{(k)} = \text{BLEU}(\mathbf{z}^{(k)}, \mathbf{y}) + \rho * q^{(k)}, \qquad (8)$$

where weight $\rho$ is controlled by an annealing schedule $\frac{e}{e+10}$ with $e$ being training epoch number. The first few epochs mainly rely on BLEU scores. The quality score is more balanced towards the evaluation score $q^{(k)}$ with increasing number of training epochs.

Therefore, the samples with high quality rate that exceed capacity $C$ are discarded (7-th line). The remaining cases are further rewriten using rewriter $\psi$ to get new translation $\mathbf{z}^{(k)}$ and quality scores

$r^{(k)}$ from Eq. (8) (11-th to 14-th lines). Eventually, we train the rewriter $\psi$ and the evaluator $\phi$ respectively using Eq. (7) and Eq. (6) (16-th and 17-th lines).

This method is designed to be both effective and efficient for training the models with *Rewriter-Evaluator* framework. For effectiveness, it keeps the low-quality translated samples in the queue for possible multiple passes of usages, facilitating the re-editing paradigm (Niehues et al., 2016) in training. For efficiency, it can obtain training time comparable to training an encoder-decoder architecture without multi-pass decoding. Assume the run-time cost for training a sample is $P$ and the training set has $Q$ batches. Using parallel mini-batch computation, the training time for encoder-decoder is about $(P * B)/B * Q = P * Q$. Analogously, the total training time for the proposed method is about $(P * (B * E))/(B * E) * Q = P * Q$.

## 4 APPLICATIONS

We apply *Rewriter-Evaluator* to improve RNNSearch and Transformer, both of which are widely adopted in NMT. In the following, we denote the length of the source sentence as $M$ and the length of translation at $k$-th pass as $L_k$.

**Improving RNNSearch.** Here both encoders $f^{SE}$ and $f^{TE}$ are Bi-directional GRU (Chung et al., 2014). We omit the details of RNNSearch and denote its hidden state at word position $i$ as $\mathbf{s}_i$ and its next word distribution on a predefined vocabulary $\mathbf{V}$ as $\pi_i^V \in R^{|V|}$.

The rewriter uses pointer generator (Gu et al., 2016; See et al., 2017) architecture. Words in the translated sentence are predicted sequentially from the predefined word vocabulary $\mathbf{V}$ or directly copied from the target candidate words. Firstly, for position $i$, pointer attention $\pi_{i,j}^S \in R^{L_{k-1}}$ is computed on the $j$-th word in the previous translation $\mathbf{z}^{(k-1)}$ as

$$\beta_{i,j} = \mathbf{v}_D^T \tanh(\mathbf{W}_D \mathbf{s}_i + \mathbf{V}_D \mathbf{p}_j^{(k-1)}), \quad \pi_{i,j}^S = \frac{\exp(\beta_{i,j})}{\sum_{1 \le j \le L_{k-1}} \exp(\beta_{i,j})}, 1 \le j \le L_{k-1}. \quad (9)$$

Then, for position $i$, its word $y_i$ is selected from the predefined vocabulary in size $|V|$ and the previous translation $\mathbf{z}^{(k-1)}$. Hence, the probability $\pi_i[y_i]$ is computed as

$$\begin{cases} \lambda_i * \pi_i^V[y_i] + (1 - \lambda_i) * \sum_{0 \le j \le L_{k-1}, \, w=\mathbf{z}_j^{(k-1)} \cap w=y_i} \pi_{i,j}^S & \text{If } y_i \in \mathbf{\Omega} \, \cap \, y_i \in \mathbf{z}^{(k-1)} \\ \pi_i^V[y_i] & \text{If } y_i \in \mathbf{\Omega} \, \cap \, y_i \notin \mathbf{z}^{(k-1)} \\ \sum_{0 \le j \le L_{k-1}, \, w=\mathbf{z}_j^{(k-1)} \cap w=y_i} \pi_{i,j}^S & \text{If } y_i \notin \mathbf{\Omega} \, \cap \, y_i \in \mathbf{z}^{(k)} \end{cases} . \quad (10)$$

where weight $\lambda_i$ is computed as $\frac{1}{1+\exp(\mathbf{u}_D^T \mathbf{h}_i)}$.

For evaluator $\phi$, in the beginning, given context representations that are computed via Eq. (2), we apply co-attention mechanism (Parikh et al., 2016) to capture the semantic alignment between element $i$ of source sequence $\mathbf{x}$ and element $j$ of translation $\mathbf{z}^{(k)}$ as

$$\begin{cases} \alpha_{i,j} = \mathbf{h}_i^T \mathbf{W}_E \mathbf{p}_j^{(k)}, 1 \le i \le M; 1 \le j \le L_k \\ \tilde{\mathbf{h}}_i = \sum_{1 \le j \le L_k} \frac{\exp(\alpha_{i,j})}{\sum_{1 \le j \le L_k} \exp(\alpha_{i,j})} \mathbf{p}_j^{(k)} \\ \tilde{\mathbf{p}}_j^{(k)} = \sum_{1 \le i \le M} \frac{\exp(\alpha_{i,j})}{\sum_{1 \le i \le M} \exp(\alpha_{i,j})} \mathbf{h}_i \end{cases} , \quad (11)$$

We then use mean pooling to extract features and compute the quality score:

$$q^{(k)} = \mathbf{v}_E^T \left( \left( \frac{1}{N} \sum_{1 \le i \le M} \tilde{\mathbf{h}}_i \right) \oplus \left( \frac{1}{L_k} \sum_{1 \le i \le L_k} \tilde{\mathbf{p}}_j^{(k)} \right) \right), \quad (12)$$

where operation $\oplus$ is the column-wise vector concatenation.

**Improving Transformer.** We keep the whole architecture (Vaswani et al., 2017) unchanged and only modify input-output format and mask matrices.

The input to Transformer is modified to $\mathbf{x}'$ as a concatenation of source sentence $\mathbf{x}$, a special symbol "ALIGN", and the last translation $\mathbf{z}^{(k-1)}$ (i.e., $\mathbf{x}' = \mathbf{x} \oplus [\text{"ALIGN"}] \oplus \mathbf{z}^{(k-1)}$). We design the following mask matrix so that words in $\mathbf{x}$ cannot attend to those in $\mathbf{z}^{(k-1)}$ and vice versa:

$$\begin{bmatrix} \mathbf{1}_{M \times M} & \mathbf{0}_{1 \times M}^T & \mathbf{0}_{M \times L_{k-1}} \\ \mathbf{1}_{1 \times M} & 1 & \mathbf{1}_{1 \times L_{k-1}} \\ \mathbf{0}_{L_{k-1} \times M} & \mathbf{0}_{1 \times L_{k-1}}^T & \mathbf{1}_{L_{k-1} \times L_{k-1}} \end{bmatrix}. \tag{13}$$

This arrangement of the input mask also enables symbol "ALIGN" to attend to all of the words in both source sentence $\mathbf{x}$ and past translation $\mathbf{z}^{(k-1)}$.

For rewriter at $k$-th pass, we adopt the same computation procedure in Eq. (9) and Eq. (10). To accomodate its sequential generation of a word at position $i$, here the hidden state $\mathbf{s}_i$ is the output from Transformer for generating target word at position $i - 1$ and $\mathbf{p}_j^{(k-1)}$ is from Transformer for position $j$ in the previous translation $\mathbf{z}^{(k-1)}$.

The evaluator obtains a score using the representation $\mathbf{h}^{ALIGN}$ from symbol "ALIGN" as

$$q^{(k)} = \mathbf{v}_E^T \mathbf{h}^{ALIGN}, \tag{14}$$

in which vector $\mathbf{v}_E$ is a learnable parameter.

## 5 EXPERIMENTS

We have conducted extensive experiments on two translation tasks, Chinese-English (Zh→En) and English-German (En→De). The results demonstrate that *Rewriter-Evaluator* significantly improves the performances of NMT models (e.g., RNNSearch) and notably outperforms prior post-editing methods. We further conducted oracle experiments, ablation studies, and running time comparison to verify the effectiveness of our framework.

### 5.1 SETTINGS

For Chinese-English translation, the training set consists of 1.25M sentence pairs extracted from LDC corpora. They are mainly chosen from LDC2002E18, LDC2003E07, LDC2003E14, Hansards portion of LDC2004T07, LDC2004T08, and LDC2005T06. We use NIST 2002 (MT02) dataset as the validation set, which has 878 sentences, and the NIST 2004 (MT04), NIST 2005 (MT05), and NIST 2006 (MT06) datasets as the test sets, which contain 1788, 1082 and 1664 sentences respectively. For English-German translation, we use WMT 2015[1] training data that contains 4.46M sentence pairs with 116.1M English words and 108.9M German words. Following previous works, we segment words via byte pair encoding (BPE) (Sennrich et al., 2015). The news-test 2013 was used as dev set and the news-test 2015 as test set. We measure the translation quality by BLEU metric (Papineni et al., 2002). For Zh→En, we adopt case-insensitive BLEU score. For En→De, case-sensitive BLEU score is calculated by *multi-bleu.pl* script[2].

We train all models with sentences of length up to 50 words. We set the vocabulary size to 30K for Chinese-English translation, and 50K for English-German translation. For RNNSearch and RNNSearch w/ *Rewriter-Evaluator*, the dimension of word embedding and hidden layer is 512, and the beam size in testing is 10. Dropout rate is set as 0.5. For Transformer and Transformer w/ *Rewriter-Evaluator*, we use default settings in fairseq (Ott et al., 2019). The batch size is fixed as 80 and the expected iterations $E$ is set as 3. The values of $\delta$ in Eq. (5) and maximum iterations $K$ are set as 0.01 and 6, respectively. We adopt *Rmsprop* (Graves, 2013).

### 5.2 MAIN RESULTS

Our models are compared with strong baselines: 1) Deliberation Network (Xia et al., 2017) adopt a second decoder for polishing the raw sequence produced by the first-pass decoder; 2) ABD-

---

[1]http://www.statmt.org/wmt15/translation-task.html.
[2]https://github.com/moses-smt/mosesdecoder/blob/master/scripts/generic/multi-bleu.perl.

| Method | | Zh→En (NIST) | | | | En→De |
|---|---|---|---|---|---|---|
| | | MT04 | MT05 | MT06 | Avg. | (WMT'15) |
| Deliberation Network (Xia et al., 2017) | | 40.56 | 37.67 | 37.20 | 38.48 | 26.01 |
| ABD-NMT (Zhang et al., 2018) | | 41.20 | 38.07 | 37.59 | 38.95 | 26.26 |
| Adaptive Multi-pass Decoder (Geng et al., 2018) | | 41.43 | 38.54 | 37.86 | 39.28 | 26.77 |
| Our Work | RNNsearch (Bahdanau et al., 2015) | 40.41 | 36.57 | 36.01 | 37.66 | 25.08 |
| | w/ *Rewriter-Evaluator* | **43.13** | **39.82** | **40.95** | **41.30** | **27.16** |
| | Transformer (Vaswani et al., 2017) | 47.11 | 47.58 | 46.70 | 47.13 | 26.45 |
| | w/ *Rewriter-Evaluator* | **48.65** | **48.31** | **49.59** | **48.85** | **28.13** |

Table 1: Experiment results of our models and all baselines. The numbers in bold indicate that the improvements are statistically significant with p-value $p < 0.05$ under t-test.

NMT (Zhang et al., 2018) utilize a backward decoder to generate a translation, and a forward decoder refines it with an attention model; and 3) Adaptive Multi-pass Decoder (Geng et al., 2018) integrate polishing mechanism into NMT model via RL.

For Zh→En translation, we adopt all the results of baselines as reported in Geng et al. (2018). For En→De, we use the results of ABD-NMT in Geng et al. (2018). Other performances are obtained by our re-implementations. For RNNSearch, Transformer, and *Rewriter-Evaluator*, we implement them on top of fairseq (Ott et al., 2019)[3].

The main results are presented in Table 1. The proposed *Rewriter-Evaluator* consistently improves RNNSearch and Transformer. Our best results are obtained by using *Rewriter-Evaluator* on Transformer. It's the first time that a NMT model with iterative rewriting consistently outperforms other post-editing based neural machine translation baselines on large translation tasks by sizeable margins. We describe the improvements in detail in the following.

**Rewriter-Evaluatoron on RNNSearch.** Improving RNNSearch with *Rewriter-Evaluator* obtains relative improvements of 9.67% on NIST dataset and 8.29% on WMT'15 dataset. These results confirm that our framework is effective in improving translation quality of encoder-decoder models (Sutskever et al., 2014; Bahdanau et al., 2015). BLEU scores of our framework with RNNSearch are also consistently higher than methods in post-edit mechanisms. For example, compared with Adaptive Multi-pass Decoder, the best one of them, we achieve higher BLEU scores by relatively 5.14% and 1.46% on NIST and WMT'15 dataset, respectively.

**Rewriter-Evaluator on Transformer.** Improving Transformer with *Rewriter-Evaluator* achieves relative improvements of 3.65% and 6.35% on NIST dataset and WMT'15 dataset, respectively. These improvements are especially significant, considering that Transformer itself consistently outperforms RNNSearch in our experiments. The progress of using *Rewriter-Evaluator* on Transform is very apparent when comparing it to the post-editing methods in Xia et al. (2017); Zhang et al. (2018); Geng et al. (2018). For example, comparing with the best post-editing method of Adaptive Multi-pass Decoder (Geng et al., 2018), *Rewriter-Evaluator* on Transformer improves BLEU scores sizeably by relative 24.36% on NIST dataset and 5.08% on WMT'15 dataset.

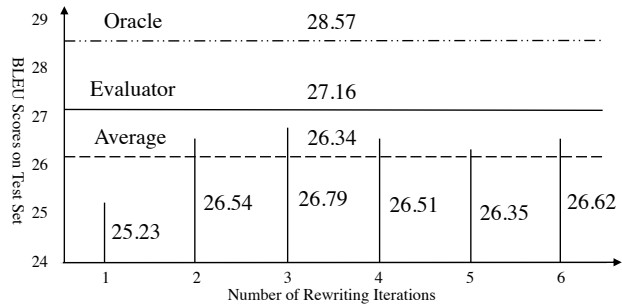

Figure 2: Oracle experiment on En→De task with *Rewriter-Evaluator* on RNNSearch.

---

[3] https://github.com/pytorch/fairseq.

### 5.3 ORACLE EXPERIMENT

One of the attractive features of our framework is its ability to select final translation results from translation candidates based on scores from evaluator. We plot the test set BLEU scores versus rewriting turn $k$ in Figure 5.2. The first iteration corresponds to the encoder-decoder result. BLEU scores from each iteration are improved over the first iteration but are consistently lower than using evaluator to select answers from them. For instance, iteration 3 obtains the highest BLEU score of 26.34 for individual iteration, but is worse than 27.16 from Evaluator. This strongly verifies the effectiveness of the evaluator. We also achieve an oracle result by selecting translations with the highest BLEU scores given ground truths. The oracle makes a higher BLEU score by 5.11%, indicating potential improvements from the further refinement of evaluator $\phi$.

### 5.4 ABLATION STUDIES

Table 2 shows ablation studies. The last row is from *Rewriter-Evaluator* on Transformer.

**Parameter Sharing.** The encoders parameters from Eq. (2) are shared in Eqs. (3) and (4) so that their representations are consistently improving for both evaluator and rewriter. Indeed, the scores from not sharing parameters, where rewriter and evaluator have their own encoders, drop by 1.32% and 3.34%.

| S | C | $\rho$ | $\delta$ | $K$ | Zh→En | En→De |
|---|---|---|---|---|---|---|
| ✗ | | | | | 48.21 | 27.19 |
| | ✗ | | | | 48.02 | 27.45 |
| | | 0.0 | | | 47.99 | 27.23 |
| | | 0.1 | | | 48.65 | 27.87 |
| | | 1.0 | | | 48.26 | 27.51 |
| | | | 0.0 | | 48.53 | 27.71 |
| | | | 0.1 | | 47.82 | 27.15 |
| | | | | 2 | 48.55 | 27.56 |
| | | | | 4 | 48.71 | 27.82 |
| | | | | 8 | **48.87** | 28.09 |
| ✓ | ✓ | A | 0.01 | 6 | 48.85 | **28.13** |

Table 2: Ablation studies. S denotes parameter sharing, C denotes copy mechanism, and A denotes weight annealing.

**Copy Mechanism in Rewriter.** Ability of copying words from past translation via Eqs. (9) and (10) contributes to 1.73% and 2.42% BLEU score increase.

**Annealing in Prioritized Training.** Equation (8) uses $\rho$ with annealing to incorporate evaluator scores. Interestingly, only using BLEU scores in selecting training samples, i.e., $\rho = 0.0$, decreases BLEU scores on test set by 1.76% and 3.20% (3-rd row). On the other hand, using fixed value (4-st and 5-st rows), the scores drop by 0.81% and 1.56%. We observe that evaluators are prone to make wrong judgment initially, therefore adopting the annealing scheme that gradually increases $\rho$ to 1.0 is indeed useful. The results also indicate the scores from evaluator contain complementary information from BLEU score.

**Threshold $\delta$.** The threshold $\delta$ in Eq. (5) controls the tolerance of stoping the iterative rewrite process on sentences that may have low translation quality. With $\delta$ set to 0.0, the Rewriter continues its editing until it cannot improve evaluation scores further. However, test set BLEU scores decline by 0.66% and 1.49% on both datasets. On the other hand, having a large tolerance, using $\delta = 0.1$, is catastrophic (7-th line) as many low-quality translation are not rewritten at all.

**Maximum Number of Iterations $K$.** Increasing the maximum number of iterations $K$ in training and test is generally beneficial. However, as it consumes more time with larger $K$ during testing, we set the maximum iterations $K$ to 6 and expected iterations $E$ to 3 in Algorithm 1.

### 5.5 RUNNING TIME COMPARISON

Table 3 shows the running times of different models. We use 8 GPUs with expected iteration number $E = 3$, maximum iteration number $K = 6$, and beam size being 5. Results confirm that training times between *Rewriter-Evaluator* and encoder-decoder are comparable.

Running time for evaluation using *Rewriter-Evaluator* is approximately 4 times of an encoder-decoder model. Nevertheless, Table 1 demonstrates that *Rewriter-Evaluator* significantly improves

| Method | Zh→En (NIST) | | En→De (WMT'15) | |
|---|---|---|---|---|
| | Training | Evaluation (ave.) | Training | Evaluation |
| RNNSearch | 2h17m | 7m | 8h12m | 12m |
| RNNSearch w/ *Rewriter-Evaluator* | 2h39m | 31m | 9h03m | 47m |
| Transformer | 1h11m | 9m | 5h47m | 14m |
| Transformer w/ *Rewriter-Evaluator* | 1h29m | 36m | 6h15m | 53m |

Table 3: Running time comparison on NIST and WMT'15.

BLEU scores in comparison to encoder-decoder models. For instance, on NIST dataset, improving RNNSearch via *Rewriter-Evaluator* increases the testing time by $4.43$ times but significantly improves the BLEU score by $8.81\%$.

## 6 RELATED WORK

Our work is closely related to recent efforts in end-to-end multi-pass decoding (Xia et al., 2017; Zhang et al., 2018; Geng et al., 2018). The models generate multiple target sentences for a source sentence and, except for the first one, each of them is based on the sentence generated in previous turn. For example, Xia et al. (2017) propose deliberation network that uses a second decoder to polish the raw sequence produced by the first-pass decoder. While these methods have achieved promising results, they lack proper termination policy to the multi-pass translation process. Zhang et al. (2018) adopt a predefined number of decoding passes, which is not flexible. Geng et al. (2018) incorporate post-editing mechanism into NMT model via RL. However, RL is notoriously unstable for training because of the high variance of gradient estimation.

An alternative line of research focuses on computer-assisted translation (CAT) (Barrachina et al., 2009) that collaborates existing machine translation technologies with human translators. In such a situation, quality estimation (QE) and automatic post-editing (APE) play important roles in reducing human efforts. Word-level QE (Kim & Lee, 2016; Martins et al., 2017; Fan et al., 2019) assigns a label of OK or BAD to every word in the translation. For example, work in Basu et al. (2018) measures the similarity of the source context of the target word with the context for which the word is retained. APE corrects typical and repetitive mistakes found in the generated sequence. For instance, work in Vu & Haffari (2018) interleaves generating and executing the edit actions to rectify errors. Most recently, some works explore multi-task learning of QE and APE (Chatterjee et al., 2018; Kim et al., 2017), which is analogous to us. While the translation quality has indeed been improved, these approaches heavily rely on extra handcraft annotation that is expensive and domain-specific. Moreover, they essentially serve as the post-process modules in a pipeline-based translation system, instead of directly empowering machine translation models with mechanisms to estimate and improve the translation quality.

## 7 CONCLUSION

In this work, we present a novel framework, *Rewriter-Evaluator*, that aims at achieving proper terminations for multi-pass decoding. It consists of a rewriter and an evaluator. At each translation pass, the rewriter generates a new translation to improve previous translations and the evaluator estimates the translation quality to determine whether to terminate the process. We also propose a prioritized gradient descent method that biases the training samples toward rewriting those low-quality translations. This enables training *Rewriter-Evaluator* that has multiple pass decodings to have comparable training time to training encoder-decoder models that only have single pass decoding. We have applied *Rewriter-Evaluator* to improve RNNSearch and Transformer. Extensive experiments have been conducted on two translation tasks, Chinese-English and English-German, showing that the proposed framework has notably improved the performances of NMT models and has significantly outperformed previous methods.

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
