# OpenReview forum: "Rewriter-Evaluator Framework for Neural Machine Translation"
_ICLR.cc/2021/Conference — Reject_

### Official Review · AnonReviewer2 · 2020-10-26
**An interesting idea but the results are not solid.**

**Rating:** 4
**Confidence:** 5

**Review:**

** Summary **
In this work, the authors proposed a rewriter-evaluator framework for neural machine translation (NMT). The translation is achieved by an iterative way: at the $k$-th iteration, a rewriter generates a translation sequence $z^{(k)}$ based on the source input $x$ and the output of previous iteration $z^{(k-1)}$. The evaluator is used to provide a score for $z^{(k)}$, used to indicate when to stop refinement.
The authors used priority queue to organize the training samples, where the sequences with low translation quality will be processed first.
The authors conduct experiments on NIST Zh->En and WMT’15 En->De.

** Pros **
(1) Proposed a new method for improving the NMT in an iterative way.

** Cons **
(1) Experimental results not convincing.
(2) The method lacks a strong explanation why it works.

** Clarify **
I think the clarify of the current version should be greatly improved.
1. What is the configuration of your Transformer and what does ``we use default settings in fairseq (Ott et al., 2019)’’ mean? Please give the specific number of layers, hidden dimensions, dropout, batchsize here. By the way, according to my knowledge, people always use Adam for optimization in NMT. Why do you choose Rmsprop?
2. Where is Figure 5.2?

** Significance **
1. About the evaluator: In Section 2, I do not get how $q^{(k)}$ is obtained in detail. It is not enough for me to get Eqn.(4) only. If you mean that the detailed definition is in Eqn.(12), I do not think it is a good measure for quality estimation, since it only measures the similarity between source input the previous translation in some hidden space. That is, the evaluator is not well-defined.
2. Experimental results not convincing. Why don’t you evaluate on WMT’14, where the baselines are provided in (Ott et al., 2019)?
3. The sacreBLEU score should also be reported (at least, in the appendix), so that we can compare your results with more algorithms in a fair way.
4. It is better to give several cases on how your method gradually improves the translation quality.


** About the baseline **
I am not satisfied with results of WMT’15 En-De as you reported in Table 1.
There are some pre-trained checkpoints on English->German, which is available at
https://github.com/pytorch/fairseq/tree/master/examples/translation.
I download the model `transformer.wmt16.en-de`, whose md5sum is (evaluated by the command line md5sum)
``68f2ea37f8eb5431f1c6b92c44c6a7bf  model.pt``

The data is preprocessed by
```
BINHOME=`the place to store mosesdecoder and subword-nmt`
SCRIPTS=$BINHOME/mosesdecoder/scripts
TOKENIZER=$SCRIPTS/tokenizer/tokenizer.perl
NORM_PUNC=$SCRIPTS/tokenizer/normalize-punctuation.perl
REM_NON_PRINT_CHAR=$SCRIPTS/tokenizer/remove-non-printing-char.perl

cat $fname | \
perl $NORM_PUNC $lang | \
perl $REM_NON_PRINT_CHAR | \
perl $TOKENIZER  -l $lang | \
python $BINHOME/subword-nmt/subword_nmt/apply_bpe.py -c $BPE_CODE  > ${fname}.bpe
```
And the translation is obtained by
```
cat wmt15.en.bpe | \
python3 $FAIRSEQ/interactive.py /tmp/wmt16.en-de.joined-dict.transformer \
--source-lang en --target-lang de \
--path /tmp/wmt16.en-de.joined-dict.transformer/model.pt \
--buffer-size 2000 --batch-size 128 --beam 5 --remove-bpe | \
grep ^H | cut -f3- > out
```
The multi-bleu/sacreBLEU for WMT’14 are 29.14/29.0, which are almost consistent with the results in (Ott et al., 2019) (i.e., 29.3 vs 28.6)
However, the multi-bleu/sacreBLEU for WMT’15 are 31.65/31.8, which are much higher than your baseline and algorithm. Therefore, I think you should re-implement your method following the settings in (Ott et al., 2019) to get more convincing results.

---

> ### Author Response · Authors · 2020-11-25
> **Response to AnonReview2**
>
> Thanks for your valuable comments. For reproducibility, we will release the source code once the paper is accepted.
>
> Commnet-1: What is the configuration of your Transformer and what does ``we use default settings in fairseq (Ott et al., 2019)’’ mean? Please give the specific number of layers, hidden dimensions, dropout, batchsize here. By the way, according to my knowledge, people always use Adam for optimization in NMT. Why do you choose Rmsprop?
>
> Answer-1: The settings of these hyper-parameters are layer=6, dimension=512, dropout=0.1, and batch=80. For fair comparisons, our experiment settings largely follow ABD-NMT [2] and Adaptive Multi-pass Decoder [3]. The two prior works both use Rmsprop for optimization.
>
> Comment-2: Where is Figure 5.2?
>
> Answer-2: It should be Figure 2. We will correct it in the revised version.
>
> Comment-3: About the evaluator: In Section 2, I do not get how q^{(k)} is obtained in detail. It is not enough for me to get Eqn.(4) only. If you mean that the detailed definition is in Eqn.(12), I do not think it is a good measure for quality estimation, since it only measures the similarity between source input the previous translation in some hidden space. That is, the evaluator is not well-defined.
>
> Answer-3:  q^{(k)} is the translation quality of the k-th translation p^k on source sentence x. Therefore, we think that the Evaluator that receives source sentence x and translation candidate p^k as the input is properly designed. Please also see the Oracle Experiment (Section 5.3) in our paper. The BLEU score of using Evaluator to select rewrite candidates is much higher than using {1,2,3,4,5,6}-th rewrites only and their average. This tells that our proposed quality score q^{(k)} indeed makes sense.
>
> Comment-4: Experimental results not convincing. Why don’t you evaluate on WMT’14, where the baselines are provided in (Ott et al., 2019)?
>
> Answer-4: The use of NIST Zh->En dataset and WMT’15 En->De dataset follows the experiment settings of previous works [1,2,3] on multi-pass decoding. We will report the results on publicly available WMT’17 Zh->En dataset in the revised version.
>
> Comment-5: The sacreBLEU score should also be reported (at least, in the appendix), so that we can compare your results with more algorithms in a fair way.
>
> Answer-5: The use of multi-BLEU follows the baselines [1,2,3]. We will report our results evaluated by sacrebleu in the revised paper.
>
> Comment-6: It is better to give several cases on how your method gradually improves the translation quality.
>
> Answer-6: We will add a case study as you require in the revised version.
>
> Comment-7: ** About the baseline ** I am not satisfied with results of WMT’15 En-De as you reported in Table 1.
>
> Answer-7: In fact, some hyper-parameters such as training epochs and model size have a great impact on the performances of models. For fair comparisons, we largely follow the experiment settings of prior baselines [1,2,3]. For example, we set the training epochs as 5 and adopt RMSProp as the optimization algorithm. Besides, almost all the results of baselines are directly copied from their papers. We will additionally report the results as you require in the revised version.
>
> [1] Xia et al., Deliberation Networks: Sequence Generation Beyond One-Pass Decoding, NIPS-2017
>
> [2] Zhang et al., Asynchronous Bidirectional Decoding for Neural Machine Translation, AAAI-2018
>
> [3] Geng et al., Adaptive Multi-pass Decoder for Neural Machine Translation, EMNLP-2018

---

### Official Review · AnonReviewer4 · 2020-10-28
**interesting idea but many questions not addressed**

**Rating:** 4
**Confidence:** 4

**Review:**

This paper proposes a rewriter-evaluator framework for multi-pass decoding, where translation hypothesis at last iteration is refined by the rewriter and further evaluated by the evaluator to decide whether to terminate the iterative decoding process. Compared to previous studies, the evaluator offers this framework the capability of flexibly controlling the termination. The authors also propose a prioritized gradient decent algorithm that biases the training process to those low-quality translation samples. Experiments on NIST Zh-En and WMT15 En-De demonstrate that the proposed model significantly outperforms several strong baselines.

Pros:

= The idea behind the rewriter-evaluator framework is easy to follow.

= The proposed method achieves significant performance improvement against several multi-pass baselines on both Zh-En and En-De translation tasks.

= The authors demonstrate that the proposed training algorithm has similar training time to the vanilla baseline, i.e. no training time loss (Table 3).

Cons:

= Some model details are missing, and the NIST Zh-En training data is not publicly available so it’s hard to exactly replicate the experiments.

= Although the framework enables flexible termination, the evaluator requires a threshold that has a large impact on translation quality and must be carefully tuned (Table 2).

= Baselines and model optimization should be further improved to fully convincing readers.

My detailed comments are as follows:

1.	The authors claim that their model can better handle the termination. One important experiment is to testify how many iterations the model uses for translating one sentence, and what factors could affect the iteration number, such as sentence length (would long inputs require more passes?). Particularly, it would be great to have an experiment to compare the termination difference between the proposed model and the Adaptive Multi-pass Decoder (Geng et al., 2018), and show evidence how the proposed one outperforms Geng et al., 2018 (i.e. RL-based model).

2.	The authors didn’t provide full model details about how to combine source encoder outputs (x) and target encoder outputs (z) in the decoder part, i.e. more details about Ea. (4) are required.

3.	Compared to other baselines, the authors adopt the copy mechanism. Could you please provide an ablation study to justify its impact, such as retraining a model using the rewriter-evaluator framework without copying?

4.	For Transformer, the authors concatenate the source input and target translation but disables cross-attention over them. Could you please give some explanation behind this practice? What if we allow the cross-attention here?

5.	Using non-public dataset, like NIST Zh-En, is not suggested in my opinion. Other researchers might not be able to replicate this experiment at all. Running experiments with (C)WMT Zh-En would be a better alternative. Besides, using tokenized BLEU with ‘multi-bleu.perl’ is also not suggested nowadays. Use sacrebleu and report the signature, instead.

6.	The results, in Table 1, for WMT15 En-De on newstest2015 are not convincing. Based on my own experience, a standard Transformer-base model can already achieve a tokenized BLEU score of ~29. I think this weak baseline comes from the fact that the authors train Transformer with batch size of 80 and optimize model parameters with RMSProp, as shown in experimental settings, Section 5.1. The authors should update their experimental training.

7.	If I understand correctly, Algorithm 1 requires online decoding process, that is, performing greedy or beam search decoding during training to get real-time estimation for q and r. In my experience, the decoding is very time-consuming. However, results in Table 3 show that there is almost no training time difference! Could you please show some theoretical explanation about this? How did you perform decoding, or get z^k from z^k-1 in practice?

If the authors can address all my concerns, I’d like to update my scores.

---

> ### Author Response · Authors · 2020-11-25
> **Response to AnonReview4 (Comments 1~5)**
>
> Thanks for your valuable feedbacks. For reproducibility, we will release the source code once the paper is accepted.
>
> Comment-1: Some model details are missing, and the NIST Zh-En training data is not publicly available so it’s hard to exactly replicate the experiments.
>
> Answer-1: We will elaborate on the model details in the revised version. The use of NIST Zh->En dataset and WMT’15 En->De dataset follows the experiment settings of previous works [1,2,3] on multi-pass decoding. We will report the performances on WMT’17 Zh->En, a publicly available dataset, in the revised version.
>
> Comment-2: Although the framework enables flexible termination, the evaluator requires a threshold that has a large impact on translation quality and must be carefully tuned
>
> Answer-2: The threshold \delta is tuned on the dev set via grid search. This is very fast because dev set is generally small.
>
> Comment-3: Baselines and model optimization should be further improved to fully convincing readers.
>
> Answer-3: Almost all the results of baselines are directly copied from their papers [1,2,3]. To make fair comparisons, our experiment setting (e.g., optimizer and training epochs) largely follows ABD-NMT [2] and Adaptive Multi-pass Decoder [3].
>
> Comment-4: The authors claim that their model can better handle the termination. One important experiment is to testify how many iterations the model uses for translating one sentence, and what factors could affect the iteration number, such as sentence length (would long inputs require more passes?). Particularly, it would be great to have an experiment to compare the termination difference between the proposed model and the Adaptive Multi-pass Decoder (Geng et al., 2018), and show evidence how the proposed one outperforms Geng et al., 2018 (i.e. RL-based model).
>
> Answer-4: The table below is the empirical distribution of rewriting iterations for the cases in the test set of NIST Zh->En. We set K = 6 and our model is Transformer w/ Rewriter-Evaluator.
>
>         1               2              3             4                5               6
>
>        5.64%           20.78%         40.37%        13.15%           10.94%          9.13%
>
> From the table, we can see that the appropriate iteration number for different examples may be different. According to your comment, we have also counted the averaged rewriting iteration number for the source sentences of different lengths. The results are shown in the table below.
>
>     1-10   11-20    21-30    31-40    >=41
>
>     1.19   2.35     3.11     3.65     4.07
>
> From the table, we can see that longer source sentences indeed incur more rewriting iterations. There may exist other causes and we will make further studies in future work. We will also design an experiment to compare the termination difference between our framework and Adaptive Multi-pass Decoder (Geng et al., 2018) in the revised paper.
>
> Comment-5: The authors didn’t provide full model details about how to combine source encoder outputs (x) and target encoder outputs (z) in the decoder part, i.e. more details about Ea. (4) are required.
>
> Answer-5: Does the reviewer mean Equation 3, instead of Equation 4 (since you have mentioned “the decoder part”)? In fact, Equation 3 offers great flexibility to design a model using our proposed framework. In Applications (Section 4), we have shown how to apply our framework to improve RNNSearch and Transformer. The Equations 9, 10, 13 in this Section are exactly the details about Equation 3.
>
> [1] Xia et al., Deliberation Networks: Sequence Generation Beyond One-Pass Decoding, NIPS-2017
>
> [2] Zhang et al., Asynchronous Bidirectional Decoding for Neural Machine Translation, AAAI-2018
>
> [3] Geng et al., Adaptive Multi-pass Decoder for Neural Machine Translation, EMNLP-2018

---

> > ### Author Response · Authors · 2020-11-25
> > **Response to AnonReview4 (Comments 6~10)**
> >
> > Comment-6: Compared to other baselines, the authors adopt the copy mechanism. Could you please provide an ablation study to justify its impact, such as retraining a model using the rewriter-evaluator framework without copying?
> >
> > Answer-6: In fact, Ablation Studies (Section 5.4) in our paper have already included the experiment you require. For fairer comparisons, we also augment the baselines with copy mechanism. The results are in the table below.
> >
> >      Method                                                          Zh->En (Avg.)                En->De
> >
> >      Deliberation Network w/ Copy Mech.                              38.72                        26.13
> >
> >      ABD-NMT w/ Copy Mech.                                           39.18                        26.37
> >
> >      Adaptive Multi-pass Decoder w/ Copy Mech.                       39.41                        26.89
> >
> >      RNNsearch w/ Rewriter-Evaluator                                 41.30                        27.16
> >
> >      Transformer w/ Rewriter-Evaluator                               48.85                        28.13
> >
> > Our models still notably outperform these improved baselines. We will add these results in the revised version.
> >
> > Comment-7: For Transformer, the authors concatenate the source input and target translation but disables cross-attention over them. Could you please give some explanation behind this practice? What if we allow the cross-attention here?
> >
> > Answer-7: The reason for this design is to keep Transformer w/ Rewriter-Evaluator consistent with RNNSearch w/ Rewriter-Evaluator, where the source sentence and the target translation are separately fed into different encoders. We also investigate a study that allows cross-attention. The results are in the table below.
> >
> >      Method                                                Zh->En (Avg.)       En->De
> >
> >      Transformer                                           47.13               26.45
> >
> >      Transformer w/ Rewriter-Evaluator                     48.85               28.13
> >
> >      Transformer w/ Rewriter-Evaluator (Cross-Attn)        48.86               28.11
> >
> > From the table, we can see that our BLEU scores change very little. We will clarify this in the revised version.
> >
> > Comment-8: Using non-public dataset, like NIST Zh-En, is not suggested in my opinion. Other researchers might not be able to replicate this experiment at all. Running experiments with (C)WMT Zh-En would be a better alternative. Besides, using tokenized BLEU with ‘multi-bleu.perl’ is also not suggested nowadays. Use sacrebleu and report the signature, instead.
> >
> > Answer-8: The use of NIST Zh->En dataset and WMT’15 En->De dataset follows the experiment settings of prior works [1,2,3] on multi-pass decoding. We will report the results on WMT’17 Zh->En in the revised version. We will also report our performances evaluated by sacrebleu in the revised paper.
> >
> > Comment-9: The results, in Table 1, for WMT15 En-De on newstest2015 are not convincing. Based on my own experience, a standard Transformer-base model can already achieve a tokenized BLEU score of ~29. I think this weak baseline comes from the fact that the authors train Transformer with batch size of 80 and optimize model parameters with RMSProp, as shown in experimental settings, Section 5.1. The authors should update their experimental training.
> >
> > Answer-9: In fact, some hyper-parameters such as training epochs and model size have a great impact on the performances of models. For fair comparisons, we largely follow the experiment settings of prior baselines [1,2,3]. For example, we set the training epochs as 5 and adopt RMSProp as the optimization algorithm. We will additionally report the results of using the recommended settings in fairseq in the revised version.
> >
> > Comment-10: If I understand correctly, Algorithm 1 requires online decoding process, that is, performing greedy or beam search decoding during training to get real-time estimation for q and r. In my experience, the decoding is very time-consuming. However, results in Table 3 show that there is almost no training time difference! Could you please show some theoretical explanation about this? How did you perform decoding, or get z^k from z^k-1 in practice?
> >
> > Answer-10: There are two key points: 1) The rewrite z^{(k)} is obtained by greedy search, which is very fast; 2) the time cost for training is, in fact, dominated by the back-propagation process. We have also adopted other tricks in engineering (e.g., torch.no_grad and parallel computation). We will make detailed comments about this in the released code.
> >
> > [1] Xia et al., Deliberation Networks: Sequence Generation Beyond One-Pass Decoding, NIPS-2017
> >
> > [2] Zhang et al., Asynchronous Bidirectional Decoding for Neural Machine Translation, AAAI-2018
> >
> > [3] Geng et al., Adaptive Multi-pass Decoder for Neural Machine Translation, EMNLP-2018

---

### Official Review · AnonReviewer3 · 2020-10-28
**Mostly solid work, but results are not fully convincing**

**Rating:** 6
**Confidence:** 3

**Review:**

In this paper, the authors complement iterative refinement for neural machine translation with an evaluator model that controls the termination of the translation process. Their approach is an alternative to the policy network used in [1].

The translation process is terminated when the predicted quality of the current translation is inferior to the previous one by a margin of $\delta$ or more, or after a fixed number of iterations otherwise. To train the models, the authors propose a sampling mechanism that prioritizes low-quality translations.

Strengths:

The proposed approach is fairly intuitive and improves upon two baselines over multiple datasets.

The paper is mostly clear and the authors give enough details to reproduce experiments, at least for En-De.

Weaknesses:

The Zh-En training data is not publicly available.

Comparison to previous work is not that convincing because most use baselines that are significantly worse than Transformer. The oracle experiment is only conducted with RNNSearch.

Questions:

In Table 3, I assume you display training time per epoch. What is the training time to convergence? Given that the sampling process is auto-regressive (i.e. conditioned on previous tokens), how can your approach have a speed comparable to standard training?

Did you train models without the priority queue, simply training over a fixed number of iterations for each example? If so, what are the results?

Why do further iterations sometimes decrease translation quality?

[1] Geng et al. Adaptive multi-pass decoder for neural machine translation. EMNLP, 2018.

---

> ### Author Response · Authors · 2020-11-25
> **Response to AnonReview3**
>
> Thanks for your valuable feedbacks. For reproducibility, we will release the source code once the paper is accepted.
>
> Comment-1: The Zh-En training data is not publicly available.
>
> Answer-1: The use of NIST Zh->En dataset and WMT’15 En->De dataset follows the experiment settings of prior works [1,2,3] on multi-pass decoding. We will report the results on WMT’17 Zh->En, a publicly available dataset, in the revised version.
>
> Comment-2: Comparison to previous work is not that convincing because most use baselines that are significantly worse than Transformer. The oracle experiment is only conducted with RNNSearch.
>
> Answer-2: In fact, all the baselines of multi-pass decoding are based on RNN. Note that our model, RNNSearch w/ Rewriter-Evaluator, also significantly outperforms all these baselines (see Table 1 in our paper). The purpose of improving Transformer with our proposed framework is to show its generality. According to your comment, we add an oracle experiment with Transformer. The results are in the table below.
>
>     Method					               En->De
>
>     1-st Rewriting				           26.17
>
>     2-nd Rewriting				           27.04
>
>     3-rd Rewriting				           27.92
>
>     4-st Rewriting				           27.13
>
>     5-st Rewriting				           27.59
>
>     6-st Rewriting				           26.83
>
>     Average					              27.11
>
>     Evaluator					            28.13
>
>     Oracle					               30.27
>
> These are consistent with the observation in the study using RNNSearch. We will add these results in our revised paper.
>
> Question-3: In Table 3, I assume you display training time per epoch. What is the training time to convergence? Given that the sampling process is auto-regressive (i.e. conditioned on previous tokens), how can your approach have a speed comparable to standard training?
>
> Answer-3: Following the experiment setting of ABD-NMT [2], we train NMT models with 5 epochs. Empirically, we find that the loss on dev set converges in 3 or 4 epochs. There are two main reasons for our efficiency: 1) The rewrite z^{(k)} is obtained by greedy search, which is very fast; 2) the running time for training is dominated by the back-propagation process. We have also applied other tricks in engineering (e.g., torch.no_grad). We will make detailed comments about this in the released source code.
>
> Question-4: Did you train models without the priority queue, simply training over a fixed number of iterations for each example? If so, what are the results?
>
> Answer-4: The table below is the empirical distribution of rewriting iterations for the cases in test set. We set K as 6. Our model is Transformer w/ Rewriter-Evaluator and dataset is NIST Zh->En.
>
>     1			2			3			4			5			6
>
>     5.64%	    20.78%       40.37%       13.15%       10.94%       9.13%
>
> These results indicate that the appropriate iteration numbers for different cases may be different. According to your comment, we conduct an experiment, where every training example is rewritten with 6 iterations. The performance on the test set is in the table below.
>
>     Method										                Zh->En (Avg.)
>
>     Transformer								 		        47.13
>
>     Transformer w/ Rewriter-Evaluator (Fixed)				48.02
>
>     Transformer w/ Rewriter-Evaluator (Adaptive)			        48.85
>
> Our BLEU score significantly declines, showing that using a priority queue to facilitate adaptive rewriting iterations is better.
>
> Question-5: Why do further iterations sometimes decrease translation quality?
>
> Answer-5: From the top table in Answer-4, we can see that the appropriate amount of rewriting iterations for different examples may be different. Therefore, over-rewriting is possible to decrease the translation quality. [3] have also observed the same phenomenon (see Table 2 in their paper).
>
> [1] Xia et al., Deliberation Networks: Sequence Generation Beyond One-Pass Decoding, NIPS-2017
>
> [2] Zhang et al., Asynchronous Bidirectional Decoding for Neural Machine Translation, AAAI-2018
>
> [3] Geng et al., Adaptive Multi-pass Decoder for Neural Machine Translation, EMNLP-2018

---

### Official Review · AnonReviewer1 · 2020-10-28
**Effective and novel multi-pass decoding for NMT**

**Rating:** 7
**Confidence:** 4

**Review:**

### Summary
This paper proposes a multi-pass generation process for NMT. It introduces an evaluator model that learns to score reference translations higher than model outputs, and serves as a policy to determine how many refinements to make during inference, and which samples to prioritize during training. Samples that have a low quality score are prioritized, such that the translation (rewriter) module is trained with a focus on low-quality inputs where rewriting can still yield improvements.
The performance of this rewriter-evaluator is evaluated on Chinese-to-English and English-to-German benchmark tasks with RNNs and Transformer architectures, and shows superior quality than previous works. Training time is reported to be comparable to a standard encoder-decoder, while inference is roughly 4 times slower. In an ablation analysis, the influence of individual components on output quality (such as a copy mechanism and hyperparameters, e.g. the maximum number of iterations) is compared. In addition, the relevance of the evaluator during inference is demonstrated in a comparison with an oracle sentence selection.

To summarize, I'd like to see this paper accepted if it addresses the questions below and makes an effort to increase reproducibility. The ideas are novel and the combination of training algorithm and architectural modifications seem effective. In seq2seq NLP applications where inference time is not a concern, the proposed methodology could improve quality in practice.

### Strengths
- The prioritized training with model outputs is novel. The idea is neat, and solves the problem of having to pick model outputs of different quality for training.
- The proposed training and models show great improvement over previous benchmarks (disregarding inference effort) on two language pairs and with two different architectures.
- The ablation shows the individual contribution of the proposed set of modifications and the performance under different hyperparameter settings.
- Training and inference time requirements are analyzed.
- The paper is written clearly and therefore easy to follow.

### Weaknesses
- Reproducibility: The code is not published, and half of the experiments are on non-freely available datasets (NIST LDC). The experiments on Zh-En could have been performed on a public benchmark like WMT 2017 as well.
- Comparison with previous works: It is unclear whether these are Transformer-based or RNN-based, and how they relate to the implementation of this paper (e.g. if the baseline MLE scores are the same), and if hyperparameter were tuned equally well. In addition, inference effort in terms of rewriting iterations is not considered in the comparison.
- Analysis: There are a few open questions (see below) that would strengthen the paper and highlight the effectiveness and the workings of the novel elements.

### Details
- Geng et al. 2018 do not report any issues with variance in their experiments, so I'd refrain from using generic RL statements to argue that their work is inferior.
- Related work in Active Learning (see below), Curriculum Learning (e.g. Zhou et al. 2020 (https://www.aclweb.org/anthology/2020.acl-main.620.pdf), and Quality Estimation (e.g. Zho et al. 2020 (https://arxiv.org/abs/2005.03519)) often use model scores directly for measuring model competence/quality/confidence, so it would be insightful to replace q with model scores to analyze if the evaluator is learning anything beyond what the rewriter already captures. See also Lee et al. 2020 (https://arxiv.org/abs/2002.07233) for relating model scores to translation quality.
- Eq 1 introduces only 2 arguments for z, but Eq. 6 feeds 3.
- The parameters are confused for the losses in Eq. 6 and 7.
- Algorithm 1: Why not directly pushing quadruples into A? There is no filter between l13 and l16.
- For the analysis, it would be great to compare how many passes each training samples receives on average during training, and how it relates to how many passes are done in inference. This would help to understand the importance of seeing model outputs during training.
- Does the baseline MT also use a pointer generator? If not, it would be good to add this for a more fair comparison.
- With longer inputs, the rewriter also performs more computation than the normal MT Transformer. Therefore, I'd like to see a comparison to a simple MT baseline where the source is concatenated twice (as described in Sec. 4) as input to the encoder. If there is no benefit from doing that, the advantage must come from inserting previous outputs.
- How are hyperparameters delta and K determined? And how large is the queue? Limiting the training to 3 epochs seems quite low.
- The comparisons to previous work are inconsistent: It is not clear which scores come from reimplementations and which ones are taken from previous papers. The reimplementation should also directly be compared to the original paper to validate the quality.
- Table 1 hides the number of iterations/rewrites. It would be great to add, to make the advantages of the proposed method shine better.
- The ablation should be done on the dev set rather than the test set.


### Related Work
- The prioritized queue is related to works in Active Learning and Curriculum Learning. Relating to these works might inspire an improvement of the algorithm or some shared insights, e.g. Liu et al. 2018 (https://www.aclweb.org/anthology/K18-1033.pdf), Platanios et al. 2019 (https://www.aclweb.org/anthology/N19-1119.pdf).

---

> ### Author Response · Authors · 2020-11-25
> **Response to AnonReview1 (Comments 1~2)**
>
> Thanks for your valuable comments. For reproducibility, we will release the source code once the paper is accepted.
>
> Comment-1: The code is not published, and half of the experiments are on non-freely available datasets (NIST LDC). The experiments on Zh-En could have been performed on a public benchmark like WMT 2017 as well.
>
> Answer-1: We will publicly release the source code once the paper is accepted. The use of NIST Zh->En dataset and WMT’15 En->De dataset follows the experiment settings of prior works [1,2,3] on multi-pass decoding. We will report the results on WMT’17 Zh->En in the revised version.
>
> Comment-2: Comparison with previous works: It is unclear whether these are Transformer-based or RNN-based, and how they relate to the implementation of this paper (e.g. if the baseline MLE scores are the same), and if hyperparameter were tuned equally well. In addition, inference effort in terms of rewriting iterations is not considered in the comparison.
>
> Answer-2: Prior works [1,2,3] on multi-pass decoding are all RNN-based. The experiments on improving Transformer is to show the generality of our proposed framework. We completely follow the experiment settings (e.g., training epochs and optimization algorithm) of prior works [1,2,3] on multi-pass decoding. We will add studies to compare the amounts of rewriting iterations between our models and the baselines in the revised version.
>
> [1] Xia et al., Deliberation Networks: Sequence Generation Beyond One-Pass Decoding, NIPS-2017
>
> [2] Zhang et al., Asynchronous Bidirectional Decoding for Neural Machine Translation, AAAI-2018
>
> [3] Geng et al., Adaptive Multi-pass Decoder for Neural Machine Translation, EMNLP-2018
>
> [4] Boyan et al., Generalization in Reinforcement Learning: Safely Approximating the Value Function, NIPS-1994
>
> [5] Anschel et al., Averaged-DQN: Variance Reduction and Stabilization for Deep Reinforcement Learning, ICML-2017
>
> [6] Mao et al., Variance Reduction for Reinforcement Learning in Input-Driven Environments, ICLR-2019

---

> > ### Author Response · Authors · 2020-11-25
> > **Response to AnonReview1 (Questions 3~8)**
> >
> > Question-3: Geng et al. 2018 do not report any issues with variance in their experiments, so I'd refrain from using generic RL statements to argue that their work is inferior.
> >
> > Answer-3: Training models with RL will be inevitably faced with the high variance of gradient estimation, since the value or state function are computed by approximation (e.g., Monte Carlo Sampling). This has been extensively discussed in RL community [4,5,6]. Our framework based on supervised learning circumvents this problem.
> >
> > Question-4: Related work in Active Learning (see below), Curriculum Learning (e.g. Zhou et al. 2020 (https://www.aclweb.org/anthology/2020.acl-main.620.pdf), and Quality Estimation (e.g. Zho et al. 2020 (https://arxiv.org/abs/2005.03519)) often use model scores directly for measuring model competence/quality/confidence, so it would be insightful to replace q with model scores to analyze if the evaluator is learning anything beyond what the rewriter already captures. See also Lee et al. 2020 (https://arxiv.org/abs/2002.07233) for relating model scores to translation quality.
> >
> > Answer-4: We have tried using the model score to replace the quality score. The results are shown in the table below.
> >
> > Method								    	 			                 Zh->En (Avg.)	      	  En->De
> >
> > RNNSearch								 			                 37.66				  25.08
> >
> > RNNSearch w/ Rewriter-Evaluator	(model score)				 39.78				  26.11
> >
> > RNNSearch w/ Rewriter-Evaluator	(quality score)			         41.30				  27.16
> >
> > Transformer								 			                 47.13				  26.45
> >
> > Transformer w/ Rewriter-Evaluator	 (model score) 	  		         47.95				  27.09
> >
> > Transformer w/ Rewriter-Evaluator	 (quality score)			         48.85				  28.13
> >
> > Using model scores degrades the performances of our models but still notably outperforms the baselines. The results indicate that the evaluators have really learned something beyond what the rewriters have already captured (exactly as the reviewer guesses). This is very interesting and we will make further exploration in future research.
> >
> > Question-5: Eq 1 introduces only 2 arguments for z, but Eq. 6 feeds 3.
> >
> > Answer-5: Equation 6 tells that the rewriter generates a sequence in an autoregressive manner. We will correct it in the revised version.
> >
> > Question-6: The parameters are confused for the losses in Eq. 6 and 7.
> >
> > Answer-6: For better understanding, we will add three terms (e.g., $\theta$, $\gamma$, and $\eta$) that denote the parameters of sentence encoders, decoder, and evaluator to Equations 6 and 7 in the revised version.
> >
> > Question-7: Why not directly pushing quadruples into A? There is no filter between l13 and l16.
> >
> > Answer-7: This makes sure that, for one training step, every sample in the priority queue A will be updated once, not zero or more. Otherwise, there may exist some samples that have not been fed into the models for many steps. We will clarify this in the revised version.
> >
> > Question-8: For the analysis, it would be great to compare how many passes each training samples receives on average during training, and how it relates to how many passes are done in inference. This would help to understand the importance of seeing model outputs during training.
> >
> > Answer-8: The two tables below are respectively the empirical distributions of rewriting iterations for the cases in training set and test set. We set E = 3 and K = 6. Our model is Transformer w/ Rewriter-Evaluator and dataset is NIST Zh->En.
> >
> > [1] Xia et al., Deliberation Networks: Sequence Generation Beyond One-Pass Decoding, NIPS-2017
> >
> > [2] Zhang et al., Asynchronous Bidirectional Decoding for Neural Machine Translation, AAAI-2018
> >
> > [3] Geng et al., Adaptive Multi-pass Decoder for Neural Machine Translation, EMNLP-2018
> >
> > [4] Boyan et al., Generalization in Reinforcement Learning: Safely Approximating the Value Function, NIPS-1994
> >
> > [5] Anschel et al., Averaged-DQN: Variance Reduction and Stabilization for Deep Reinforcement Learning, ICML-2017
> >
> > [6] Mao et al., Variance Reduction for Reinforcement Learning in Input-Driven Environments, ICLR-2019

---

> > > ### Author Response · Authors · 2020-11-25
> > > **Response to AnonReview1 (Questions 9~14)**
> > >
> > > Question-9: Does the baseline MT also use a pointer generator? If not, it would be good to add this for a more fair comparison.
> > >
> > > Answer-9: According to your comment, we have incorporated copy mechanism into the baselines. The results are in the table below.
> > >
> > > Method										                Zh->En (Avg.)			En->De
> > >
> > > Deliberation Network w/ Copy Mech.					38.72					26.13
> > >
> > > ABD-NMT w/ Copy Mech.							        39.18					26.37
> > >
> > > Adaptive Multi-pass Decoder w/ Copy Mech.			39.41					26.89
> > >
> > > RNNsearch w/ Rewriter-Evaluator						41.30					27.16
> > >
> > > Transformer w/ Rewriter-Evaluator						48.85					28.13
> > >
> > > The baselines are slightly improved but still significantly underperform our models. This confirms that our improvements are mainly contributed by the rewriting process. We will add these results in the revised version.
> > >
> > > Question-10: With longer inputs, the rewriter also performs more computation than the normal MT Transformer. Therefore, I'd like to see a comparison to a simple MT baseline where the source is concatenated twice (as described in Sec. 4) as input to the encoder. If there is no benefit from doing that, the advantage must come from inserting previous outputs.
> > >
> > > Answer-10: Based on your comment, we have tested the performances of doubling the source to a Transformer. The results are in the table below.
> > >
> > > Method									        Zh->En (Avg.)			En->De
> > >
> > > Transformer								        47.13					26.45
> > >
> > > Transformer w/ Double Source				47.14					26.41
> > >
> > > Transformer w/ Rewriter-Evaluator				48.85					28.13
> > >
> > > The BLEU scores change very little. This indicates that the improvements obtained by our models are from inserting previous outputs.
> > >
> > > Question-11: How are hyperparameters delta and K determined? And how large is the queue? Limiting the training to 3 epochs seems quite low.
> > >
> > > Answer-11: We use grid search to determine the values of \delta and K (see the ablation experiments in Section 5.4). The size of priority queue A is set as B \times E (B is batch size and E is the number of expected rewriting iterations). Following the experiment setting of ABD-NMT [2], our training epochs is 5.
> > >
> > > Question-12: The comparisons to previous work are inconsistent: It is not clear which scores come from reimplementations and which ones are taken from previous papers. The reimplementation should also directly be compared to the original paper to validate the quality.
> > >
> > > Answer-12: In Table 1, most of the performances of baselines are directly copied from their papers [1,2,3]. Only the results of Deliberation Network [1] and Adaptive Multi-pass Decoder [3] on WMT’17 Zh->En are from our re-implementations. We have checked the BLEU scores of our reproduced baselines on NIST Zh->En, which are consistent with those reported in their papers.
> > >
> > > Question-13: Table 1 hides the number of iterations/rewrites. It would be great to add, to make the advantages of the proposed method shine better.
> > >
> > > Answer-13: We will add the iteration numbers to Table 1 in the revised version.
> > >
> > > Question-14: The ablation should be done on the dev set rather than the test set.
> > >
> > > Answer-14: We will report the results on dev set rather than test set in Ablation Studies (Section 5.4) in the revised paper.
> > >
> > > [1] Xia et al., Deliberation Networks: Sequence Generation Beyond One-Pass Decoding, NIPS-2017
> > >
> > > [2] Zhang et al., Asynchronous Bidirectional Decoding for Neural Machine Translation, AAAI-2018
> > >
> > > [3] Geng et al., Adaptive Multi-pass Decoder for Neural Machine Translation, EMNLP-2018
> > >
> > > [4] Boyan et al., Generalization in Reinforcement Learning: Safely Approximating the Value Function, NIPS-1994
> > >
> > > [5] Anschel et al., Averaged-DQN: Variance Reduction and Stabilization for Deep Reinforcement Learning, ICML-2017
> > >
> > > [6] Mao et al., Variance Reduction for Reinforcement Learning in Input-Driven Environments, ICLR-2019

---

### Decision · Program_Chairs · 2021-01-07
**Final Decision**

**Decision:**

Reject

**Comment:**

This paper builds upon recent iterative refinement approaches NMT with an evaluator model that controls the termination of the translation process, yielding a “rewriter-evaluator framework” for multi-pass decoding. Their approach is an alternative to the policy network used in Geng et al (EMNLP 2018). The main delta wrt previous studies is that the evaluator offers this framework the capability of flexibly controlling the termination. While the idea behind the rewriter-evaluator framework is sensible and well described, and the proposed method achieves significant performance improvement against reported baselines, reviewers pointed out some concerns with the baselines and model optimization details. More analysis of the termination procedure against the RL-based model of Geng et al. 2018 could shed some light on why the proposed approach is better. Some analysis testifying how many iterations the model uses for translating one sentence, and what factors could affect the iteration number, such as sentence length, would greatly improve the paper. A second weakness pointed out by reviewers is related to the results of WMT’15 En-De reported in Table 1, where the reported baseline numbers seem to be weaker than expected. As pointed out by one the reviewers, pre-trained checkpoints on English->German (available at https://github.com/pytorch/fairseq/tree/master/examples/translation) exist which achieve much higher sacre BLEU than the reported baseline. I found the authors’ answer not very convincing regarding this point. Therefore, I recommend rejection. I suggest the authors, in future iterations of their work, address some of the issues pointed out by the reviewers and re-implement their method following the settings in (Ott et al., 2019) to get more convincing results.